# Beyond Accuracy: Toward Trustworthy Tabular Foundation Models in Industrial Applications

Johannes von Keler [1]  Jan Achterhold [1]  Luiz Ricardo Douat [1]  Maria Lyssenko [1]  Mark Schillinger [1]
Matthias Woehrle [1]

## Abstract

Tabular foundation models (TabFMs) show immense promise on academic benchmarks, but their transition to real-world industrial applications requires more than just predictive accuracy. In these high-stakes environments, properties like reliability, robustness, and ultimately, the trust of engineers, are paramount. We argue that to leverage these advanced models successfully, the evaluation focus must broaden from isolated accuracy metrics to a more holistic, application-centric view of model behavior.

To illustrate what such an evaluation could entail, we present a targeted study of TabFMs on two industrial engine datasets. We showcase how performance is affected by key factors such as the number of estimators and the training data fraction. Furthermore, on controlled synthetic functions, we analyze the models' uncertainty quantification in the presence of noise and their behavior under distribution shift, probing their interpolation versus extrapolation capabilities. Our investigations demand for model cards for TabFMs, TabFM-enabled data and model analysis routines, and holistic benchmarking for reliable and trustworthy application of TabFMs on industrial prediction tasks.

## 1. Introduction

Tabular foundation models such as TabPFN (Hollmann et al., 2022; 2025), TabICL (Qu et al., 2025; 2026) and others (Ma et al., 2025; Bouadi et al., 2025) promise several benefits for tabular regression and classification tasks (van Breugel & van der Schaar, 2024). They drastically outperform classical algorithms in benchmarks (Erickson et al.,

[1]Robert Bosch GmbH, Stuttgart, Germany. Correspondence to: Johannes von Keler <Johannes.vonKeler@bosch.com>.

*Proceedings of the $2^{nd}$ ICML Workshop on Foundation Models for Structured Data*, Seoul, South Korea. 2026. Copyright 2026 by the author(s).

2025), are remarkably easy to use for practitioners, and provide (marginal) predictive posterior distributions, making them interesting for uncertainty-informed downstream tasks such as Bayesian optimization (Yu et al., 2025) and active learning (Margraf et al., 2024).

For a trustworthy application in industrial prediction tasks, however, tabular foundation models not only introduce additional capabilities, but also challenges. In this paper, we outline three directions to increase trust in industrial prediction tasks with tabular foundation models. *First*, we argue that the combination of low entry barriers to using TabFMs, the wide variety of available models, and the implicit assumption in designing those demands the introduction of *model cards for tabular foundation models*, in analogy to model cards for large language models (Mitchell et al., 2019). Exemplarily, the data used in pretraining shapes the predictive posterior distributions, which should be made explicit. We outline potential properties for such model cards and experimentally investigate the behaviour of TabPFN and TabICL under these properties. *Second*, we show that TabFMs, due to providing uncertainty estimates, allow for computing metrics on data and model ablations that go beyond predictive accuracy on pre-defined splits. This, in turn, allows for creating *data and model ablation analysis routines* for increased robustness and trustworthiness. *Last*, we argue, that these new capabilities demand for a *holistic benchmarking* of tabular prediction methods, similar to what Liang et al. (2023) propose for Large Language Models. In our experiments, we provide more details about metrics these benchmarks should, from our standpoint, contain, such as predictive accuracy for low data regimes and evaluation of uncertainty calibration (Landsgesell & Knoll, 2026b).

## 2. Background and Related Work

A promising direction in tabular data modeling is the development of foundation models that perform In-Context Learning (ICL). Primarily based on the Transformer architecture, these models process a new task's training data as part of their input prompt to make predictions, bypassing the need for training or fine-tuning. Pioneering examples, such as the Prior-Fitted Network (TabPFN) (Hollmann et al.,

2022; 2025) and TabICL (Qu et al., 2025; 2026), are pretrained on purely synthetic data to learn a general-purpose inference algorithm. While there are various further models (Spinaci et al., 2025; Zhang et al., 2025b;a) that differ in pretraining such as using real data and support for additional features such as semantic interpretation of text fields, we focus on the two most commonly used models, which additionally performed best in our experiments. The focus of all of these works has been overwhelmingly on the model itself, i.e., its architecture, pre-training strategy, and achieving state-of-the-art predictive accuracy on academic benchmarks.

A critical aspect of building industrial trust is moving beyond simple accuracy to properly quantify model uncertainty. This need is highlighted by recent work from Landsgesell and Knoll, who argue that simple metrics are insufficient where misprediction costs are asymmetric (Landsgesell & Knoll, 2026a;b). Building on principles of probabilistic forecasting, they assert that evaluating the full posterior predictive distribution—that is, assessing not just *what* the model predicts, but also how *confident* it is in that prediction—with proper scoring rules is essential for creating reliable and well-calibrated models (Gneiting & Raftery, 2007). The introduction of their ScoringBench benchmark facilitates this robust, application-aware assessment, an approach this work builds upon (Landsgesell & Knoll, 2026b).

## 3. Towards Trustworthy Application of Tabular Foundation Models in Industry

**Model cards for TabFMs**   In classical algorithms for prediction and classification tasks, e.g. Gaussian processes (Rasmussen & Williams, 2006), assumptions (e.g., the prior over functions) are explicit. In tabular foundation models, these assumptions are more implicit–embedded in the synthetic data generation scheme during pretraining. We argue that these assumptions should be made transparent via *model cards* (Mitchell et al., 2019), as they are critical for the correct use and the interpretation of results, particularly predictive posteriors. Exemplarily, a model card should address: Which classes of functions served as empirical priors and with which frequency were they sampled? Shall categorical variables be represented one-hot or integer? How does the model handle missing data? Can column names be included? While answers to some of these questions can be found in the respective publications, consolidating them into a standardized model card would greatly improve accessibility. This becomes especially relevant as tabular foundation model based prediction algorithms reach a broader audience who may not yet be familiar with these subtleties.

**New capabilities for data and model analysis**   Industrial deployment requires trust in both predictions and their uncertainty. Since TabFMs provide posterior distributions out of the box, we can move beyond point-accuracy evaluation toward uncertainty-aware analysis. In our experiments, we demonstrate metrics for accuracy-(Figure 2) and uncertainty-based (Figure 3) split selection.

**Holistic benchmarks**   New capabilities bring new evaluation challenges. Current tabular benchmarks focus on prediction accuracy (RMSE, $R^2$), but TabFMs demand broader assessment: posterior quality, low-data regime performance, and sensitivity to distribution shift (Sec. 4).

These dimensions are interconnected: noise calibration on synthetic data benchmarks uncertainty quality *and* reveals pretraining assumptions for model cards; hyperparameter sensitivity analysis belongs in both benchmarks and documentation. Holistic evaluation frameworks exist for language (Liang et al., 2023), code (Jain et al., 2025), audio (Yang et al., 2025), and video (Lee et al., 2024) foundation models—but not yet for tabular. We advocate for similar efforts in this domain.

The following experiments are designed to illustrate these relationships within an industrial context and provide hints for essential information and validation tests required for comprehensive model cards.

## 4. Experiments

We investigate the behavior of two tabular foundation models, TabPFNv2.6 (Grinsztajn et al., 2025) and TabICLv2 (Qu et al., 2026) with respect to a non-exhaustive set of properties and evaluations we see as important for characterizing model behavior, data analysis, and holistic benchmarking.

**General Setup**   For real-world experiments, we use the Bosch Engine Dataset (Longi et al., 2022): two combustion engines Engine 1 and Engine 2, four input features, seven regression targets (*cf.* Sec.A in the appendix). This dataset was chosen as it represents a common engineering scenario, characterized by a small sample size and a regression task. We focus on three representative targets: HC (emissions), TIC (thermal), and ERS (roughness). Among these, ERS and HC are consistently challenging across both engines, while TIC is comparatively easy to predict.

Further, we split each engine dataset into a train (80%) and test (20%) split unless otherwise noted. Leave-One-Out (LOO) analysis confirms TabICL, TabPFN, and GP achieve comparable accuracy (Section B.1). We report coefficient of determination $R^2 = 1 - \sum_i (y_i - \hat{y}_i)^2 / \sum_i (y_i - \bar{y})^2$, root mean squared error $\text{RMSE} = \sqrt{\frac{1}{n} \sum_i (y_i - \hat{y}_i)^2}$, and continuous ranked probability score $\text{CRPS} = \mathbb{E}[|Y - y|] -$

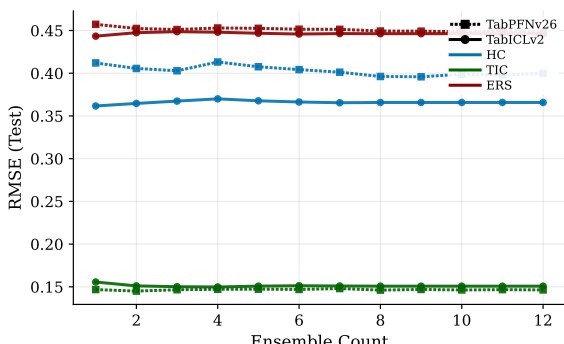

*Figure 1.* **How sensitive is the prediction accuracy regarding the $M$ (n_estimators) hyperparameter?** RMSE stabilizes by $M \approx 8$; additional estimators yield no accuracy gain.

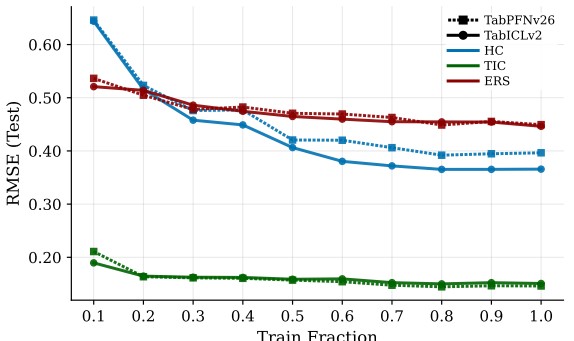

*Figure 2.* **How sensitive is the prediction accuracy w.r.t. reduced training sets?** This probes low-data capabilities: RMSE declines with a linear trend throughout the training fraction range.

$\frac{1}{2}\mathbb{E}[|Y - Y'|]$ on the test split, where $Y, Y'$ are independent draws from the predictive distribution.

**Ensemble sweep**  We fix the training data and vary the number of ensemble members $M \in \{1, \ldots, 12\}$. Each member applies different internal transformations[1], and predictions are averaged as $\hat{y}_i = \frac{1}{M}\sum_m \hat{y}_i^{(m)}$. This isolates the effect of ensemble size from data variability.

**Training fraction sweep**  We fix $M = 8$ and vary the fraction $f \in \{0.1, \ldots, 0.9, 1.0\}$ of training data used. This evaluates sample efficiency—how both point accuracy (RMSE) and calibration (CRPS) scale with data availability.

We analyze both engines but focus on Engine 2 here—Engine 1 exhibits flat or quickly saturating behavior, offering less diagnostic insight.

**Scalability analysis: ensemble size and sample efficiency**
Each sweep experiment uses an independent random test

---

[1]Each ensemble member uses its own internally sampled transformations; these transformations are part of the model's internal procedure and are not explicitly specified by us.

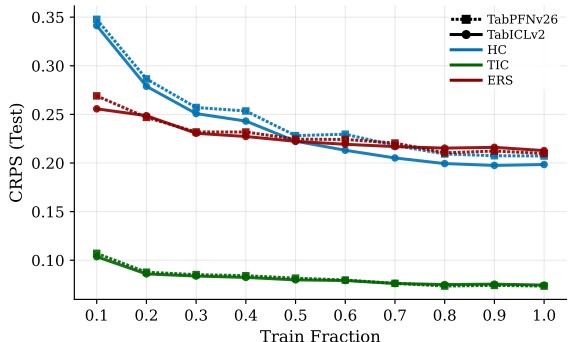

*Figure 3.* **How sensitive is the uncertainty calibration w.r.t reduced training sets?** This confirms that calibration benefits from additional data in tandem with point accuracy; both RMSE and CRPS decline approximately linearly.

split. While absolute metric values are not directly comparable across experiments, our analysis focuses on *trends*—how metrics evolve as a function of the swept parameter.

For the Engine 2 dataset (Figure 1), increasing the number of estimators yields no accuracy gains. A plausible reason for this could be the dataset's limited feature count and small sample size. If this pattern holds true for small datasets in general, practitioners could leverage this behavior to achieve faster inference, particularly in recursive modeling scenarios.

For the training fraction (Figures 2 and 3), Engine 2 shows similar decreases in both RMSE and CRPS throughout the range. HC and ERS remain the most challenging targets. This consistency across models indicates a *data-centric* effect.

Holistic benchmarking across training fractions reveals target-specific data requirements and where collection efforts saturate. Crucially, high point accuracy can mask poor calibration–joint RMSE and CRPS evaluation ensures predictions are both accurate and trustworthy for deployment.

**Influence of noise level on RMSE and prediction uncertainty**  To assess how tabular foundation models adapt their predictive uncertainty to noise, we conduct a controlled experiment on synthetic data: a quadratic bowl function $f(x)$ on a 2D input space. The first $n_{\text{test}} = 1000$ points of a Sobol sequence serve as test data; from the subsequent 1000 samples we use subsets $n \in \{100, \ldots, 1000\}$ as training data. While samples from a Sobol sequence are not i.i.d., this is a standard approach for DoEs which maximizes space-filling uniformity. Training targets are corrupted with additive Gaussian noise $y = f(x) + \epsilon$, where $\epsilon \sim \mathcal{N}(0, \sigma^2)$ for $\sigma \in \{0, 0.5, 2.0\}$. The test set remains noiseless.

As expected, the RMSE decreases with the amount of training data and increases with higher noise levels (*cf.* Fig-

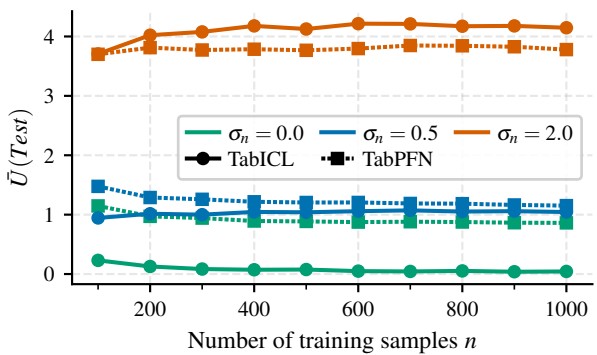

*Figure 4.* **Can TabFMs recover synthetic aleatoric noise?** A comparison of TabPFN's and TabICL's mean predictive uncertainty $\bar{U}$ on the quadratic bowl benchmark function. For well-calibrated uncertainties, $\bar{U}$ should approach $2\sigma$, a behavior we observe for TabICL but not always for TabPFN.

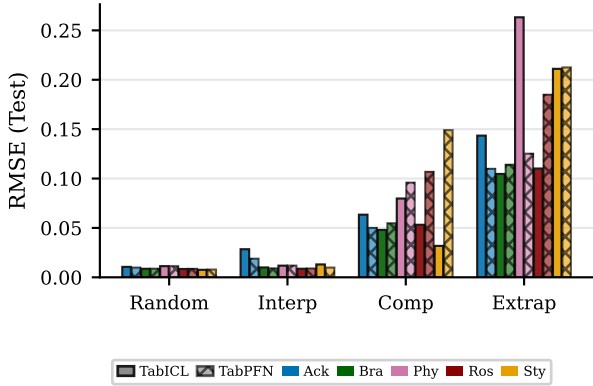

*Figure 5.* **How do TabFMs behave in inter- and extrapolation regimes?** We artificially create feature subsets, similar to (Schott et al., 2022), to assess various inter- and extrapolation settings on 2D synthetic data (Ackley, Branin, Physics, Rosenbrock, Physics). More details in sec. 4.

ure 10). TabPFN and TabICL perform comparably on this dataset.

Figure 4 shows that both models adapt their uncertainty to noise level. We define the mean prediction uncertainty $\bar{U} = \frac{1}{n_{\text{test}}} \sum_{i=1}^{n_{\text{test}}} \left( q_{0.84}(x_i) - q_{0.16}(x_i) \right)$. For well-calibrated predictions, $\bar{U}$ should approach $2\sigma$, since 68% of samples from $\mathcal{N}(\mu, \sigma^2)$ lie within $\mu \pm \sigma$. All models roughly meet this expectation, except TabPFN on noiseless data, where it is underconfident ($\bar{U} \to 1$ instead of 0). Notably, $\bar{U}$ stabilizes after 100–200 samples, while RMSE continues decreasing until 300–1000 samples depending on noise level.

We conclude that tabular foundation models estimate data noise well, even with limited training data—a critical property for establishing trust in predictions and their uncertainty bounds in industrial applications. We further analyze differences across the tabular foundation models in Section B.4.

**Controlled validation on synthetic benchmarks**  To enhance transparency, we argue that model cards and holistic benchmarks should document robustness under distribution shift. As one example of generalization testing within holistic evaluation, we adapt a rigorous protocol from computer vision (Schott et al., 2022) to controlled synthetic benchmarks. Our analysis uses a simplified 2D setup: while success here does not guarantee generalization in higher dimensions, failure provides a strong signal of potential issues. We probe five known functions under four data-split protocols–random, composition, interpolation, and extrapolation–to assess in-distribution versus out-of-distribution behavior. Details are provided in Section B.2.

As shown in Figure 5, both TabICL and TabPFN exhibit consistent behavior across all five synthetic targets: *random* and *interpolation* splits are handled easily (RMSE < 0.03), confirming strong in-distribution performance. *Composi-*

*tion*—where feature values appear individually but their combination is novel—presents moderate difficulty. *Extrapolation* is most challenging, revealing robustness limits when test inputs lie outside the training distribution. This difficulty hierarchy (random ≈ interpolation < composition < extrapolation) is model-agnostic, reflecting a data-centric challenge rather than architectural limitations.

## 5. Conclusion

We argue that while tabular foundation models drastically simplify prediction tasks for the practitioner, additional scrutiny has to be taken to ensure the trustworthiness of the results. This is mainly due to assumptions during the design and training of such models. For a practitioner, those are less obvious compared to "classical" prediction algorithms. A prominent example is the implicit prior over functions which shape the posterior predictive probabilities. With this work, we aim to draw attention of the community towards embedding TabFMs into (*i*) consensus for a model card for tabular foundation models to expatiate internals of the models and allow a practitioner to correctly invoke and interpret the results of a TabFM, (*ii*) prediction systems based on TabFMs supporting the practitioner in ensuring trustworthiness of the results, and (*iii*) the need for holistic benchmarks quantifying the capabilities of models with respect to various metrics needed to enable such prediction systems.

**Acknowledgment**  We thank Prior Labs for granting us permission to use TabPFN 2.6 in this evaluation and the anonymous reviewer for their insightful comments.

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

# A. Engine Calibration Datasets

## A.1. Overview

We evaluate on two real-world engine calibration datasets from the publicly available Bosch Engine Datasets repository (Longi et al., 2022). The *pengines* collection provides steady-state measurements of the *same* combustion engine recorded under two different operating conditions (denoted **Engine 1** and **Engine 2**). We use these datasets because they closely resemble the structure and scale of internal calibration data encountered in practice.

Each dataset consists of $N$ input–output measurement pairs $\{(\mathbf{x}_n, \mathbf{y}_n)\}_{n=1}^{N}$, with all columns independently standardized to zero mean and unit variance.

The dataset was previously used for safe active learning using multi-output Gaussian processes (Li et al., 2022).

## A.2. Structure

*Table 1.* Dataset characteristics

| Name | Abbreviation | Engine 1 range | Engine 2 range |
|---|---|---|---|
| *Inputs ($d = 4$)* | | | |
| Engine speed | | $[-1.56,\ 2.85]$ | $[-1.52,\ 3.44]$ |
| Engine load | | $[-2.01,\ 2.01]$ | $[-2.05,\ 2.21]$ |
| Intake valve opening | IVO | $[-1.82,\ 1.60]$ | $[-2.11,\ 1.16]$ |
| Air–fuel ratio | AFR | $[-2.85,\ 2.49]$ | $[-2.45,\ 2.70]$ |
| *Outputs ($p = 7$)* | | | |
| Specific fuel consumption | SFC | $[-1.46,\ 2.94]$ | $[-1.58,\ 2.76]$ |
| Exhaust manifold temperature | TEM | $[-1.96,\ 2.11]$ | $[-2.01,\ 2.08]$ |
| Catalyst temperature | TIC | $[-2.18,\ 2.03]$ | $[-2.01,\ 2.00]$ |
| Engine roughness – variance | ERV | $[-1.29,\ 3.46]$ | $[-1.38,\ 4.55]$ |
| Engine roughness – standard deviation | ERS | $[-1.47,\ 4.62]$ | $[-1.23,\ 4.41]$ |
| HC emissions | HC | $[-1.72,\ 2.33]$ | $[-1.70,\ 2.42]$ |
| $NO_x$ emissions | $NO_x$ | $[-2.76,\ 1.69]$ | $[-2.69,\ 1.66]$ |

*Table 2.* Summary statistics for the two engine datasets (standardized space).

| | Engine 1 | Engine 2 |
|---|---|---|
| Samples ($N$) | 795 | 785 |
| Inputs ($d$) | 4 | 4 |
| Outputs ($p$) | 7 | 7 |
| *Input operating range (standardized)* | | |
| Engine speed | $[-1.56,\ 2.85]$ | $[-1.52,\ 3.44]$ |
| Engine load | $[-2.01,\ 2.01]$ | $[-2.05,\ 2.21]$ |
| IVO | $[-1.82,\ 1.60]$ | $[-2.11,\ 1.16]$ |
| AFR | $[-2.85,\ 2.49]$ | $[-2.45,\ 2.70]$ |
| *Selected output skewness* | | |
| SFC | 0.88 | 0.81 |
| ERS | 1.15 | 1.49 |
| $NO_x$ | $-0.53$ | $-0.55$ |

## A.3. Dataset Characteristics

**Input structure.** The four inputs are largely uncorrelated (maximum pairwise Pearson $|r| \approx 0.30$ for engine speed – engine load in Engine 1, 0.20 in Engine 2). IVO is nearly orthogonal to all other inputs ($|r| < 0.03$ in both datasets), making it a relatively independent control dimension. The inputs span similar standardized ranges in both datasets, with Engine 2 exhibiting a notably wider engine speed range (max $3.44$ vs. $2.85$ std) and a narrower IVO range (max $1.16$ vs. $1.60$ std).

**Output structure.** Temperatures (exhaust manifold and catalyst) are approximately symmetric ($|\text{skew}| < 0.1$). Roughness outputs are right-skewed in both datasets (skew $\approx 1.2$–$1.5$), reflecting the occasional high-roughness operating points. $NO_x$

is mildly left-skewed ($\approx -0.54$). Engine speed is the dominant linear predictor of exhaust temperature ($r \approx 0.85$), while engine load is the dominant linear predictor of SFC ($r \approx -0.83$), consistent with physical intuition. The distributions of the outputs are depicted in the histograms Figure 6 and Figure 7.

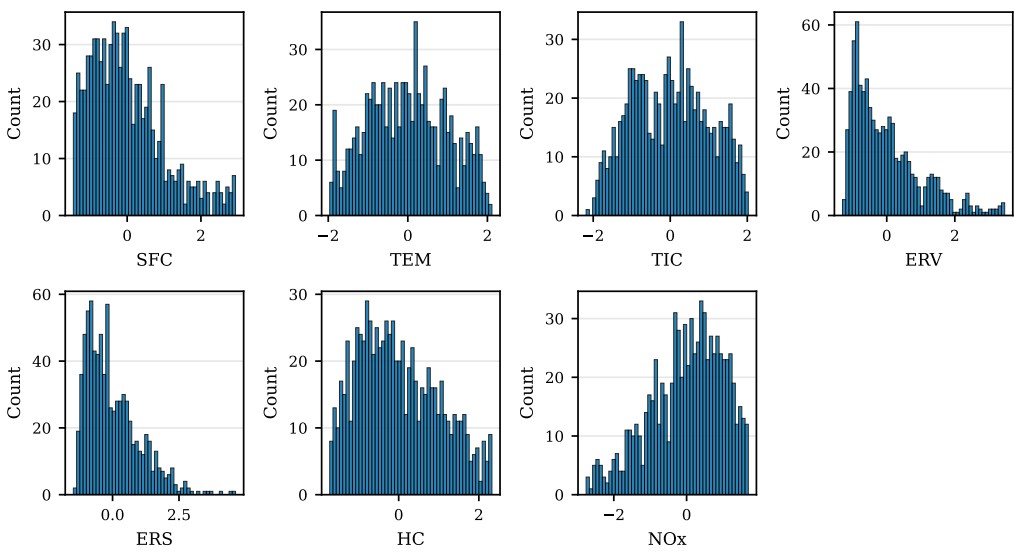

*Figure 6.* Histograms of Engine 1 outputs for each target

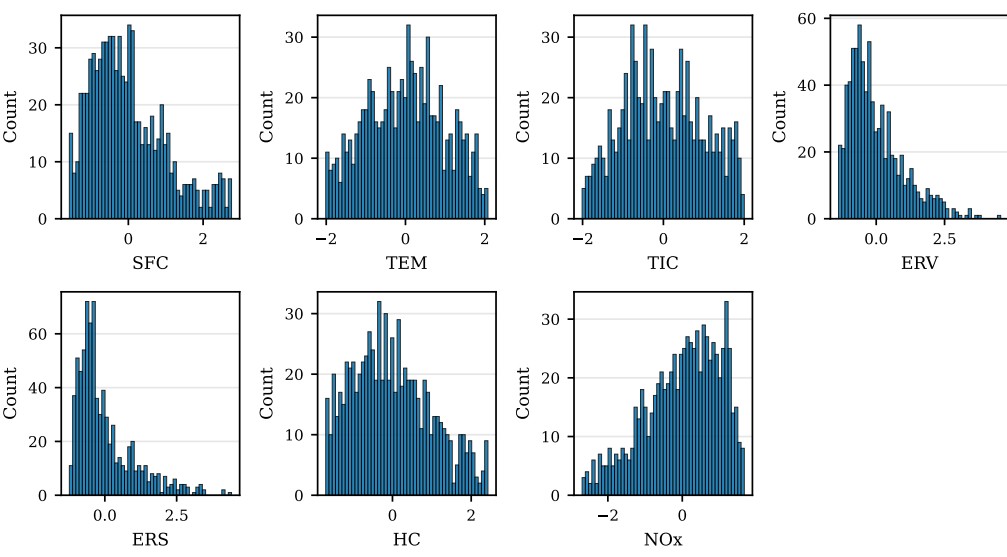

*Figure 7.* Histograms of Engine 2 outputs for each target

**Between-dataset differences.** Although both datasets share the same schema and operating system, their marginal input distributions differ beyond a simple re-scaling, making naive pooling or direct transfer non-trivial. The engine roughness outputs show increased tail mass in Engine 2 (skew $= 1.49$ vs. $1.15$ for `ERS`), suggesting more frequent extreme-roughness conditions.

## B. Experiments

We always use the default settings for both tabular foundation models if not stated differently, e.g., we deviate from the default number of estimators only in the experiment ensemble sweep in Paragraph 4.

*Table 3.* Comparison of model performance metrics across different targets of the Engine 2 dataset. Test metrics are calculated using Leave-one-out cross-validation (LOO). The best metric for each target is printed in bold face.

| Target | Model | Train RMSE | Train R² | LOO RMSE | LOO R² |
|--------|-------|-----------|----------|----------|--------|
| SFC | ASC GP | 0.1229 | 0.9849 | 0.1335 | 0.9821 |
| SFC | TabICL | 0.0842 | 0.9929 | **0.1211** | **0.9853** |
| SFC | TabPFN | **0.0792** | **0.9937** | 0.1258 | 0.9841 |
| TEM | ASC GP | 0.2372 | 0.9436 | 0.2623 | 0.9311 |
| TEM | TabICL | 0.1618 | 0.9738 | **0.2460** | **0.9394** |
| TEM | TabPFN | **0.1570** | **0.9753** | 0.2492 | 0.9378 |
| TIC | ASC GP | 0.1528 | 0.9766 | 0.1678 | 0.9718 |
| TIC | TabICL | 0.1068 | 0.9886 | **0.1547** | **0.9760** |
| TIC | TabPFN | **0.0990** | **0.9902** | 0.1580 | 0.9750 |
| ERV | ASC GP | 0.5411 | 0.7068 | 0.5962 | 0.6441 |
| ERV | TabICL | 0.3055 | 0.9066 | 0.5444 | 0.7032 |
| ERV | TabPFN | **0.2630** | **0.9307** | **0.5376** | **0.7106** |
| ERS | ASC GP | 0.4494 | 0.7978 | 0.4871 | 0.7624 |
| ERS | TabICL | 0.2655 | 0.9294 | 0.4472 | 0.7997 |
| ERS | TabPFN | **0.1932** | **0.9626** | **0.4418** | **0.8046** |
| HC | ASC GP | 0.4561 | 0.7917 | 0.5210 | 0.7282 |
| HC | TabICL | 0.1782 | 0.9682 | **0.4050** | **0.8358** |
| HC | TabPFN | **0.1571** | **0.9753** | 0.4061 | 0.8348 |
| NOx | ASC GP | 0.1825 | 0.9666 | 0.2194 | 0.9518 |
| NOx | TabICL | 0.1120 | 0.9874 | 0.2085 | 0.9565 |
| NOx | TabPFN | **0.1081** | **0.9883** | 0.2042 | 0.9582 |

### B.1. Prediction accuracy

We compare RMSE and R² of TabICL, TabPFN and an ASC GP model trained using ETAS ASCMO (Buchner et al., 2020) on the Engine 2 dataset. All models were evaluated using Leave-one-out cross-validation (LOO). As shown in Table 3, TabPFN consistently achieves the best performance on the training data across all targets. However, in the LOO evaluation, TabICL shows superior performance on four of the seven targets (SFC, TEM, TIC, and HC). The performance of the two foundation models is generally very similar. The ASC GP model's performance is lower by a small margin, but being effectively on par from an application standpoint for most targets while offering the advantage of rapid LOO computation.

### B.2. Controlled validation on synthetic benchmarks - detailed experiment description

We construct a 2D synthetic benchmark to evaluate generalization under controlled distribution shifts. We select four test functions from the global optimization literature (see Silagadze (2004); Jamil & Yang (2013) for an overview) and construct a physics-related test function.

- **Ackley** (Ackley, 1987): multimodal function with many local minima

$$f_{\text{Ackley}}(\mathbf{x}) = -a \exp\left(-b\sqrt{\frac{1}{d}\sum_{i=1}^{d} x_i^2}\right) - \exp\left(\frac{1}{d}\sum_{i=1}^{d}\cos(cx_i)\right) + a + e \tag{1}$$

with $\mathbf{x} \in [-5, 5]^2$ (i.e., $d = 2$), $a = 20$, $b = 0.2$, $c = 2$.

- **Rosenbrock** (Rosenbrock, 1960): narrow curved valley

$$f_{\text{Rosenbrock}}(x, y) = (a - x)^2 + b(y - x^2)^2 \tag{2}$$

with $(x, y) \in [-2, 2]^2$, $a = 1$, and $b = 100$.

- **Branin RCOS** (Branin, 1972): three global minima

$$f_{\text{Branin}}(x, y) = a\left(y - bx^2 + cx - r\right)^2 + s(1 - t)\cos(x) + s \tag{3}$$

with $x \in [-5, 10]$, $y \in [0, 15]$, $a = 1$, $b = \frac{5.1}{4\pi^2}$, $c = \frac{5}{\pi}$, $r = 6$, $s = 10$, and $t = \frac{1}{8\pi}$.

- **Styblinski-Tang** (Styblinski & Tang, 1990): multiple local minima

$$f_{\text{Styblinski-Tang}}(\mathbf{x}) = \frac{1}{2} \sum_{i=1}^{d} \left( x_i^4 - 16x_i^2 + 5x_i \right) \tag{4}$$

with $\mathbf{x} \in [-5, 5]^2$ (i.e., $d = 2$).

- **Physics**: projectile range ($v_0$: velocity, $\theta$: launch angle)

$$f_{\text{Projectile}}(v_0, \theta) = \frac{v_0^2 \sin(2\theta)}{g} \tag{5}$$

with $v_0 \in [5, 20]$, $\theta \in \left[0, \frac{\pi}{2}\right]$, and $g = 9.81$ the gravitational acceleration.

Inputs are drawn uniformly from $[0, 1]^2$ and mapped to each function's natural domain. Targets are MinMax-scaled to $[0, 1]$ using training data only, ensuring RMSE comparability across functions. We cannot determine if these synthetic functions were part of the pretraining due to the lack of public information about the priors. However, judging from the results a posteriori, there is strong evidence that they were not.

**Split Protocols.**  We evaluate four train/test splits ($\approx$75%/25%):

- **Random**: standard i.i.d. split

- **Composition**: test where $x_1 > 0.5$ and $x_2 > 0.5$ (individual feature values seen; their combination unseen)

- **Interpolation**: grid pattern (8×8; alternating test cells)

- **Extrapolation**: $x_1 > 0.75$ completely unseen during training

These splits probe different generalization regimes: in-distribution (random), combinatorial generalization (composition), spatial interpolation, and out-of-distribution extrapolation.

We visualize the construction of the splits in Fig. 8 below.

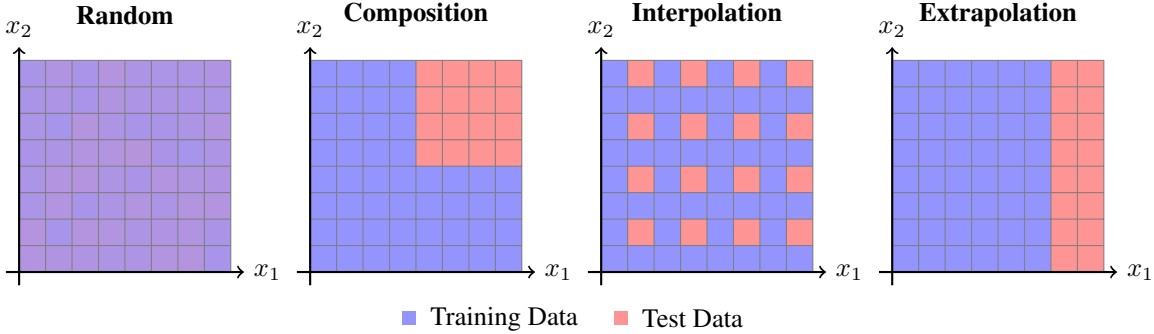

*Figure 8.* Visualization of the four train/test split strategies on an $8 \times 8$ grid. Each cell colour is derived from the data split: pure blue = all training, pure red = all test, and blended cells reflect the observed test fraction (e.g. the *Random* panel shows ˜25 % test uniformly across all cells).

### B.3. Interpolation vs. Extrapolation on Real Data

To validate the interpolation vs. extrapolation findings of the synthetic benchmark on real data, we conduct an extrapolation experiment on the Bosch Engine Dataset (Engine 2). We compare model robustness under extrapolation against a random sampling baseline to quantify how prediction quality degrades when test points fall outside the training distribution.

On real data where inputs are higher-dimensional and not uniformly distributed as in the synthetic case, we use a simplified setup comparing two splitting strategies with 70% of the data in train and 30% test:

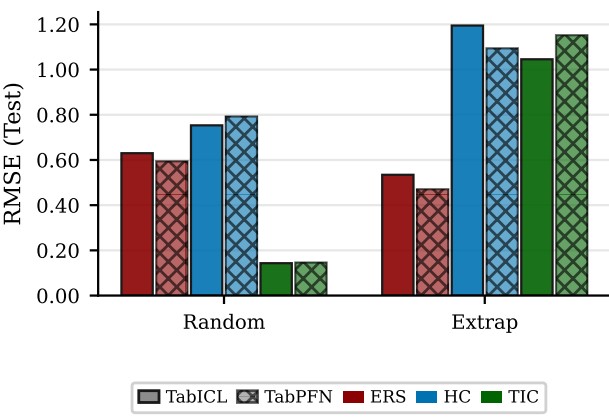

*Figure 9.* **How do TabFMs behave in inter- and extrapolation regimes?** RMSE under random vs. marginal-tail extrapolation splits on Engine 2. Extrapolation degrades HC and TIC but not ERS, suggesting target-dependent sensitivity to distribution shift.

- **Random (baseline):** Standard random shuffled splits where test points are distributed throughout the feature space, representing an interpolation regime.

- **Extrap (extrapolation on marginal tails):** For each feature dimension, samples are sorted and either the top or bottom tail is held out as the test set. This evaluates prediction quality when individual input dimensions fall outside the training range.

Figure 9 reveals a more nuanced picture than the synthetic benchmarks. For HC and TIC, extrapolation yields higher RMSE than random splitting, consistent with the expected degradation under distribution shift. However, for ERS the two strategies perform comparably, with extrapolation even showing slightly better results. This target-dependent behavior prevents a definitive ranking of random versus extrapolation performance on real data.

Interestingly, under extrapolation HC and TIC achieve similar RMSE—despite the training fraction experiments (Section 4) showing TIC significantly outperforming HC in both RMSE and calibration (CRPS) under standard splits. This suggests that the relative difficulty of targets can shift depending on the evaluation regime.

The robustness of ERS to extrapolation may relate to its distributional properties: histogram analysis in Figure 7 reveals ERS has a skewed distribution, which could make tail-based splits less extreme relative to the training distribution. Alternatively, uncontrolled noise sources in the real-world measurement process may obscure the clean interpolation–extrapolation distinction observed on synthetic data. These findings highlight that distribution shift effects on real industrial data are target-dependent and require careful per-task evaluation.

This extrapolation analysis complements the accuracy and calibration experiments to provide a holistic assessment of model behavior under varied data split construction regimes on real data.

### B.4. Influence of noise level on RMSE and prediction uncertainty – detailed experiment description

We use $f(x) = (10x_1 - 5)^2 + (10x_2 - 5)^2$, $x = (x_1, x_2) \in [0, 1]^2$ as the quadratic bowl function. Histograms visualizing the error distribution can be found in Figure 11. Linking the observed error distribution of TabPFN for $\sigma_n = 0.0$ with the mean predictive uncertainty in Figure 4, one can infer that TabPFN's predictions contain an inherent noise level. This is likely an effect of model pretraining, e.g. how training data is generated and data preprocessing. This observation would be noteworthy for the model card, as it directly helps practitioners better understand the model's outputs and builds confidence in its behavior.

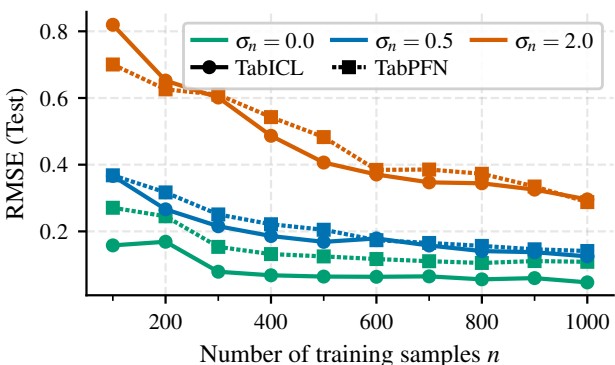

*Figure 10.* A comparison of TabPFN's and TabICL's RMSE on the quadratic bowl benchmark function depending on the amount of training samples $n$ for different noise levels.

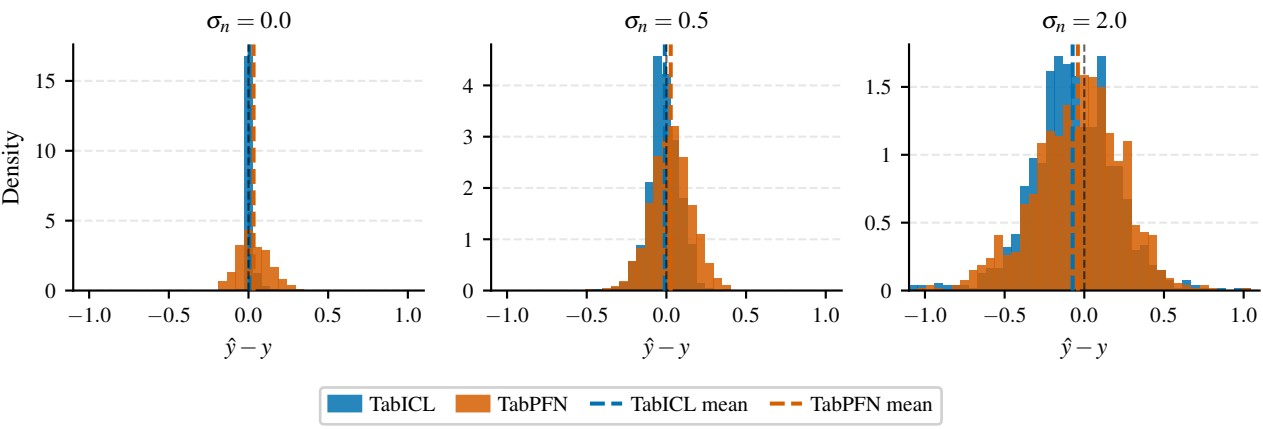

*Figure 11.* Histogram of errors on the quadratic bowl function for TabICL and TabPFN with varying noise level and $n = 1000$ training points.

