# OpenReview forum: "Beyond Accuracy: Toward Trustworthy Tabular Foundation Models in Industrial Applications"
_ICML.cc/2026/Workshop/FMSD — FMSD @ ICML 2026 Poster_

### Official Review · Reviewer_83CQ · 2026-05-18
**Review for Beyond Accuracy: Toward Trustworthy Tabular Foundation Models in Industrial Applications**

**Rating:** 6
**Confidence:** 3

**Review:**

## Summary
This paper examines the extent to which Tabular Foundation Models can be used to make reliable predictions in an industrial context. For doing so, they formulate criteria Tabular Foundation Models should fulfill like model cards, measurement of prediction trustworthiness and holistic benchmarks. Empirically, two Tabular Foundation Models are evaluated under varying conditions: on a real-world engine dataset and on synthetic data.

---

## Strengths

- it is relevant to understand how TabFMs behave in real-world settings and under varying conditions
- the 3 criteria for the usage of TabFMs are understandable and important
- experiments cover different important scenarios such as performance under data availability, noise, interpolation and extrapolation
- detailed appendix

---

## Areas for Improvement

- the paper type is not clear: it reads in the introduction and conclusion like a position/perspective paper on what is needed (future) for trustworthy applications. The experiments, on the other hand, imply that the paper is already itself a benchmark of the TabFM's on different scenarios. Also the experiments appear to be quite well executed from a technical standpoint, but they do not directly support the individual hypotheses or at least there is no interpretation related to the three requirements.
- the framing itself could be better on for whom the paper is relevant: if practitioners are targeted (what the title implies) then the experiments ask a bit the wrong questions. For a practitioner it doesn't help to know that 70% of train fraction could be enough for a required CRPS. They want to know how many samples (in absolute numbers) are required for their given data with a specific distribution to use these models. E.g. what characteristics (which distributions etc) do HC and ERS have that they benefit most from more data compared to TIC. The data and analysis is in principle in the paper for doing so (appendix) but these are the relevant points for a practitioner and should be discussed in the main paper.
- it should be at least motivated why the chosen real and synthetic datasets are representive or relevant for industrial applications.
- Baseline is missing in many cases. Without baseline (e.g. some automl which may be comparable easy to deploy as TabFM's) the benefits of TabFM's are not clear and it should be included even (or even more) if the focus is on an holistic TabFM evaluation.
- Figure 1 implies that in 4 out of 6 cases it is actually best to have only 1 estimator but it is constantly stated that M=8 ist most stable. The oscillation is not visible in the chosen visualization
- missed clarification and investigation that synthetic test data was not part of the pretraining data of TabFMs
- the sobol sequence sampling of the quadratic bowl function violates iid assumptions, which I don't see as dealbreaker as the assumptions are often violated in practice, but one should be aware
- probably typo in ntest of the quadratic bowl function experiment: ntest should probably be 100 and not 1000.


---

## Detailed Comments

Improvements can be made in connecting hypothesis and experiments with stronger focus on target audience.

---

## Justification of Score

While there are some drawbacks in the framing of the paper and in the chosen experimental designs, there are still interesting points worth discussing in the community.

---

### Official Review · Reviewer_F1si · 2026-05-19
**testing TabICL and TabPFN?**

**Rating:** 3
**Confidence:** 4

**Review:**

Summary:
"A critical aspect of building industrial trust is moving beyond simple accuracy to properly quantify model uncertainty." This is well-known. The paper advocates for the introduction/use of model cards for tabular foundation models, in analogy to model cards for large  language models

Strengths:
- The desire to codify the use of tabular foundation models with model cards and benchmarks

Areas for Improvement:
- Make the objective of the paper clearer for the reader. The paper seems to have many different objectives
- Lack of novelty/originality, more like a discussion
- About the benchmarks you advocate for, something like TabArena? https://huggingface.co/spaces/TabArena/leaderboard